# Impact of Exercise Intensity on Cerebral BDNF Levels: Role of FNDC5/Irisin

**DOI:** 10.3390/ijms25021213

**Published:** 2024-01-19

**Authors:** Clémence Leger, Aurore Quirié, Alexandre Méloux, Estelle Fontanier, Rémi Chaney, Christelle Basset, Stéphanie Lemaire, Philippe Garnier, Anne Prigent-Tessier

**Affiliations:** 1Inserm Unité Mixte de Recherche 1093-Cognition, Action & Plasticité Sensorimotrice, Unité de Formation et de Recherche Sciences de Santé, Université de Bourgogne, F-21000 Dijon, France; clemence.leger@u-bourgogne.fr (C.L.); aurore.quirie@u-bourgogne.fr (A.Q.); alexandre.meloux@u-bourgogne.fr (A.M.); estelle.fontanier@u-bourgogne.fr (E.F.); chaney-52@hotmail.fr (R.C.); christelle.basset@u-bourgogne.fr (C.B.); pgarnier@u-bourgogne.fr (P.G.); 2Centre Hospitalier Universitaire Dijon, Service de Biochimie Spécialisée, F-21000 Dijon, France; stephanie.lemaire@u-bourgogne.fr; 3Département Génie Biologique, Institut Universitaire et Technologique, F-21000 Dijon, France

**Keywords:** FNDC5, irisin, BDNF, physical exercise intensity, muscle fibers, hippocampus

## Abstract

The positive effects of physical exercise (EX) are well known to be mediated by cerebral BDNF (brain-derived neurotrophic factor), a neurotrophin involved in learning and memory, the expression of which could be induced by circulating irisin, a peptide derived from Fibronectin type III domain-containing protein 5 (FNDC5) produced by skeletal muscle contraction. While the influence of EX modalities on cerebral BDNF expression was characterized, their effect on muscle FNDC5/Irisin expression and circulating irisin levels remains to be explored. The present study involved Wistar rats divided into four experimental groups: sedentary (SED), low- (40% of maximal aerobic speed, MAS), intermediate- (50% of MAS) and high- (70% of MAS) intensities of treadmill EX (30 min/day, 7 days). Soleus (SOL) versus gastrocnemius (GAS) FNDC5 and hippocampal BDNF expressions were evaluated by Western blotting. Additionally, muscular FNDC5/Irisin localization and serum/hippocampal irisin levels were studied by immunofluorescence and ELISA, respectively. Our findings revealed that (1) serum irisin and hippocampal BDNF levels vary with EX intensity, showing a threshold intensity at 50% of MAS; (2) hippocampal BDNF levels positively correlate with serum irisin but not with hippocampal FNDC5/Irisin; and (3) GAS, in response to EX intensity, overexpresses FNDC5/Irisin in type II muscle fibers. Altogether, peripheral FNDC5/Irisin levels likely explain EX-dependent hippocampal BDNF expression.

## 1. Introduction

The regular practice of physical exercise (EX) is now well considered as the main preventive factor against many pathological processes such as cardiovascular diseases [1] as well as neurodegenerative pathologies which are associated with cognitive disorders and declines. Focusing on brain health, EX positively impacts cerebral plasticity, which improves cognitive performances in terms of learning or memorization, but also boosts psychological well-being [2,3,4]. Widely studied for its involvement in various aspects of brain function and mental health, brain-derived neurotrophic factor (BDNF) appears to be the mediator of EX’s beneficial effects in promoting neurogenesis, synaptic plasticity and cell survival, resulting in the improvement of cognitive abilities and facilitating learning processes and memory [5,6,7,8,9,10].

However, the mechanisms underlying how EX increases cerebral BDNF levels are still being studied, and currently, three distinct pathways emerge as key contributors. The first pathway involves the hemodynamic response to EX, in which an increase in shear stress on endothelial cells lining blood vessels leads to the NO-dependent synthesis of BDNF [10]. The second pathway is the neuronal pathway, in which the elevation of BDNF levels is dependent on neuronal activity [11]. Finally, the third pathway recently identified [12] involves the secretion into the bloodstream of myokines by skeletal muscles during contraction, potentially influencing brain health and indicating the existence of muscle/brain crosstalk [13,14]. The relative contribution of these mechanisms to the beneficial effects of EX-induced cerebral BDNF overproduction remains complex, and further investigations are underway to fully elucidate these interplays. Among the humoral factors shown to be secreted in response to EX, the myokine irisin, discovered in 2012 [15], has received considerable attention in the field of EX physiology. Irisin, found in the bloodstream, is derived from the cleavage of the fibronectin type III protein domain containing 5 (FNDC5), a transmembrane protein that is under the control of the peroxisome proliferator-activated receptor γ coactivator-1α (PGC-1α), a transcription factor induced in skeletal muscle in response to its contraction [15,16,17]. Although some studies suggest the ability of irisin to cross the blood–brain barrier (BBB), the mechanisms and extent of its transport are not yet fully understood. Recent work by Islam et al. (2021), using mice injected intravenously with adeno-associated virus-irisin, demonstrated that peripherally produced irisin could cross the BBB. Indeed, they showed that three weeks post-injection, irisin protein levels in the brain were elevated, while only the liver showed irisin mRNA expression among other peripheral organs [18]. It has been hypothesized that FNDC5/Irisin could lead to beneficial cerebral effects through cerebral BDNF overproduction [12]. Moreover, administered at the onset of a mouse model of middle cerebral artery occlusion (MCAO), recombinant irisin (intracerebroventricular injection) has been shown to attenuate brain damage by reducing apoptosis and increasing BDNF protein in the ischemic brain cortex [19]. Thereby, irisin could serve as a potential mediator between muscle and brain in pathophysiological conditions. In addition to its peripheral expression, FNDC5/Irisin is also produced in different brain regions. Wrann et al. (2013) reported that EX increases cerebral BDNF expression through the PGC-1α/FNDC5 pathway, while the knockdown of PGC-1α reduced FNDC5 expression in the brain [12]. In a previous study conducted in rats [20], we demonstrated that brain BDNF overexpression was dependent on EX intensity following treadmill EX. Building upon this finding and considering the potential interaction between irisin and cerebral BDNF levels during EX, our present study was designed to investigate the impact of different EX intensities on serum irisin levels and to identify the primary cellular source of irisin found in the blood of young adult rats.

## 2. Results

### 2.1. Baseline Characteristics of Sedentary and Exercised Rats

At the end of the MAS test, animals were allocated to four different groups of six rats each: sedentary (SED), low- (12 m/min, EX12), medium- (14 m/min, EX14) and high- (18 m/min, EX18) intensity EX. As observed in Table 1, when the four groups were compared to each other, no variation in body and muscle (SOL and GAS) weights were observed, except for the SOL in EX14 group, whose weight was significantly increased compared to the SED group (*p* < 0.05). Moreover, the SOL weight (SOLW)/body weight (BW) and GAS weight (GASW)/BW ratios were not modified by EX protocol, regardless of the intensity of the EX. Similarly, the MAS incremental test average was not significantly different between the four groups.

### 2.2. Exercise Intensity Affects Hippocampal BDNF Expression

Although the increase in brain BDNF levels induced by EX is well established, to understand the role of FNDC5/Irisin in the muscle/brain crosstalk, the impact of different intensities of the EX protocol was studied on BDNF levels in the hippocampus (HP), a structure involved in learning and memory. Our previously published results showed that the EX-induced increase in BDNF was dependent on EX intensity modalities in rats subjected to low- (12 m/min, ~40% MAS) and high- (18 m/min, ~70% MAS) EX intensities on horizontal treadmill (30 min/day, 7 consecutive days) [20]. In the present study, a third group of exercised rats was added, EX14, which corresponds to rats subjected to a moderate intensity EX (14 m/min, ~50% MAS). Thus, the expression of BDNF was evaluated in the four experimental groups by Western blotting. Our results showed that all modalities of EX intensity induced an increase in hippocampal BDNF levels (Figure 1), although it was significant only for EX14 (+48.1 ± 51.8%, *p* < 0.05, Figure 1b) and EX18 (+32.1 ± 15.6%, *p* < 0.01, Figure 1c) when compared to the SED group (EX12, +33.2 ± 67.3%, ns, Figure 1a).

### 2.3. Exercise Intensity Affects Serum Irisin Levels That Correlate with Hippocampal BDNF Expression

As previously mentioned, irisin derived from FNDC5 cleavage appears to impact cerebral functions [18]. Thus, we evaluated the impact of our modalities of EX on serum irisin levels and determined whether these levels were correlated with the expression of BDNF in the HP. Of note, the serum was obtained just after the last treadmill boot, whereas the collection of other samples was carried out 24 h after.

#### 2.3.1. Serum Irisin Levels

As observed for hippocampal BDNF, EX intensity similarly affects serum irisin levels (Figure 2a). Indeed, increases of +20.5% (*p* < 0.01) and +17.0% (*p* < 0.05) were observed for EX14 and EX18, respectively, as compared to the SED group, while no significant change was noticed for EX12 despite a slight increase of +8.0% (ns). When the EX-intensity groups were compared to each other, no difference was revealed.

#### 2.3.2. Correlation of Hippocampal BDNF Expression with Serum Irisin Levels

To explore the association between circulating irisin levels and hippocampal BDNF expression in SED and EX groups, a Spearman correlation test was performed. The results showed a linear positive correlation between these two variables (Figure 2b, r_s_ = 0.5087, *p* < 0.05), suggesting that irisin secreted into the circulation from skeletal muscle during EX could induce hippocampal BDNF expression.

### 2.4. Exercise Intensity Does Not Impact Either Hippocampal FNDC5 Expression nor Irisin Levels Which Do Not Correlate with BDNF Expression in Hippocampus

To provide arguments concerning the muscle/brain crosstalk and, more particularly, the peripheral origin of cerebral BDNF overexpression, we investigated the impact of EX intensities on FNDC5 expression evaluated by Western blotting and irisin levels measured by ELISA in the HP since a hippocampal production of FNDC5/Irisin has already been described [21].

As shown in Figure 3, regardless of the EX-intensity (EX12, EX14 or EX18), no upregulation in FNDC5 expression and irisin levels was observed in the HP (Figure 3a,d). Moreover, no correlation was found when BDNF expression and irisin levels obtained in the HP were plotted together in SED and EX (EX12, EX14 and EX18) rats (Figure 3e). All these data suggest that cerebral FNDC5/Irisin may not be involved in the BDNF hippocampal overexpression following EX.

### 2.5. Exercise Intensity Impacts FNDC5/Irisin Expression and Localization in Muscles Depending on the Type of Fiber Metabolism

Next, we investigated the effect of EX intensity on FNDC5/Irisin in two types of skeletal muscles based on the metabolic activity: the SOL with oxidative metabolism mainly composed of type I slow-twitch fibers (87%) and the GAS with mixed oxidative-glycolytic metabolism mainly composed of type II (IIa and IIb) fast-twitch fibers (88%) [22].

Before determining the impact of EX modalities on FNDC5/Irisin expression/localization in the two skeletal muscles mentioned above, we first compared the expression of FNDC5 in SOL versus GAS under basal conditions (SED group). The results (Figure 4) showed that FNDC5 was strongly expressed in GAS compared to SOL muscles within the same group of animals (+7000%, *p* = 0.0313).

#### 2.5.1. FNDC5 Expression in SOL Muscle after Different Extents of EX

Analysis of FNDC5/Irisin expression in SOL at different EX intensities showed no significant variation associated with EX intensity as compared to the SED group. 

Using immunofluorescence analysis, our data reported that FNDC5/Irisin staining in SOL was predominantly expressed by type II (MyHC II) compared to type I (MyHC I) fibers (Figure 5a). In addition, type II fibers FNDC5/Irisin staining was unchanged regardless of the EX group studied when compared to the SED group (Figure 5b,c).

#### 2.5.2. FNDC5 Expression in GAS Muscle after Different Extents of EX

Conversely to SOL muscle, a significant increase in FNDC5/Irisin expression in the GAS muscle was observed from the intensity threshold of 14m/min. Indeed, moderate (EX14) and elevated (EX18) EX intensities enhanced FNDC5/Irisin expression when compared to SED (+218.9 ± 69.7% and +366.2 ± 133.0%, respectively, *p* < 0.001, Figure 6b,c), while a slight but non-significant increase was observed for EX12 (Figure 6a).

As shown for SOL, immunofluorescence analysis (Figure 7) revealed that FNDC5/Irisin staining was mostly present in predominant type II fibers (fast-twitch, Figure 7a), as in SOL, with a staining that was proportional to EX intensity (Figure 7b,c). As already observed in Western blotting analysis, FNDC5/Irisin staining was gradually increased compared to SED but significant only for EX14 (*p* < 0.05, Figure 7b) and EX18 (*p* < 0.001, Figure 7c). These results suggest that an intensity greater than 14 m/min is necessary to overexpress this myokine.

### 2.6. Correlation of GAS FNDC5/Irisin Expression with Serum Irisin Levels

To investigate the possible source of circulating irisin, correlations between muscle (SOL and GAS) FNDC5/Irisin expression and serum irisin levels were carried out. It should be noted that the correlation between muscular FNDC5 and serum irisin levels has limitations due to the temporal gap between irisin measurement (0 h) and FNDC5 expression assessment (24 h). However, we can assume that muscular FNDC5 levels measured 24 h after the last session of EX are still a good indicator of its levels post-EX, as already described by Pang et al. (2018) who showed that the levels of FNDC5 protein were elevated 1 h post-EX and maintained until 24 h [23] after one boot of EX. In addition, according to the protocol used in our study (a daily 30 min EX during 7 consecutive days), it is very likely that FNDC5 expression was already increased at the end of the seventh EX session. While no association was found in SOL, a positive correlation was observed in GAS (r_s_ = 0.6609, *p* < 0.001, Figure 8), suggesting that oxidative/glycolytic muscles, mainly composed of type II fibers, could be an important source of serum irisin levels both in sedentary and exercised conditions.

## 3. Discussion

As stated in the introduction, the mechanism underlying EX-induced cerebral BDNF expression is a complex process involving increases in neuronal activity and cerebral blood flow as well as the activation of the humoral pathway. As we previously reported an effect of EX intensity on brain BDNF production, we sought to investigate in this study whether this experimental paradigm had repercussions on the humoral pathway, focusing on the capacity of muscles to express FNDC5 and secrete irisin. To better characterize muscle and fiber types behind the FNDC5/Irisin pathway, differential expression and staining of this myokine were assessed in two muscles that differ in their metabolism: the soleus (SOL, oxidative metabolism) and the gastrocnemius (GAS, mixed oxidative-glycolytic metabolism). The main results of this study are that (i) irisin was secreted into the bloodstream in response to EX with an intensity threshold of 50% of MAS, (ii) intensity-dependent increase in hippocampal BDNF was correlated with circulating irisin levels, (iii) FNDC5 expression was preponderant in GAS compared to SOL, (iv) only GAS responded to EX intensity with a significant and gradual increase in FNDC5 staining that was mainly attributed to type II fibers, and this expression was correlated to circulating irisin levels.

The crucial role of BDNF in the positive effects of EX on the brain [24,25,26] is well established, as in animal studies, anti-BDNF strategies reversed the cognitive benefits associated with EX [5,27,28], while in humans, the val66met polymorphism associated with a defect in activity-dependent regulated BDNF secretion, attenuated the cognitive benefits of EX [29,30,31]. While neuronal activity via calcium influx and neurotransmitter release and hemodynamic pathway through elevation of cerebral blood flow have long been considered as dominant mechanisms to increase cerebral BDNF expression [32], the humoral/endocrine pathway has recently been reported to play a considerable role in cerebral BDNF production in both rodents and humans [12], highlighting muscle FNDC5 production and irisin secretion [15].

As previously reported [20], our current data showed that hippocampal BDNF expression was dependent on EX intensity, since its expression raised significance from a threshold intensity of 50% MAS. This result is also in line with a study showing a better improvement in spatial memory, hippocampal BDNF expression and neurogenesis in rats subjected to 4 weeks of high-intensity interval training (HIIT) compared to moderate-intensity continuous training (MICT) (30 min/day, 5 days/week) [33]. Similarly, a study comparing MICT and HIIT showed a progressive increase in brain BDNF expression with EX intensity (6-week protocol with 6 sessions/week) [34]. Interestingly, our data also reported that serum irisin levels followed the same variation pattern as hippocampal BDNF expression, with a significant increase from 50% of MAS. Indeed, it has already been reported that serum irisin levels were higher after HIIT than MICT with 5 sessions/week for 8 weeks [35] or with 5 sessions/week for 6 weeks [36]. Furthermore, when circulating irisin levels were plotted against hippocampal BDNF expression, a significant positive association was found between these two biological parameters. As EX has been shown to activate FNDC5/Irisin in the brain [12], we also assessed the impact of EX on hippocampal FNDC5 expression. In our experimental setup, we were unable to observe increases in FNDC5 expression and irisin levels in the hippocampus (HP). Taken together, these results strongly suggest that hippocampal BDNF overexpression is associated with peripheral FNDC5 rather than the central FNDC5 pathway, as previously suggested [37]. Although not investigated in our study and still largely unknown, our data raise the question about the mechanisms through which peripheral irisin leads to increased brain BDNF production. Despite some reports indicating that irisin could cross the blood–brain barrier (BBB) using AAV mediated overexpression of liver-derived irisin strategy [18], our data, showing that the rise in circulating levels of irisin was not accompanied by a corresponding increase in brain levels, suggest that irisin may act through an alternative mechanism. Consequently, our findings imply that peripheral irisin might communicate at the BBB, potentially through αVβ5-type integrin receptor signaling, as reported by the same group [38,39]. Considering the demonstrated expression of these receptors by the cerebral endothelium [40] and the capacity of irisin’s ability to induce a vasodilatory effect dependent on nitric oxide (NO) [41], it is reasonable to hypothesize that irisin may initially signal at the endothelium through NO-related mechanisms, subsequently inducing brain BDNF synthesis. However, this hypothesis requires further investigations.

To gain a deeper understanding of the mechanisms involved in muscle/brain crosstalk, we analyzed two muscles, the SOL and the GAS, as described above. Despite their common location in the calf muscle, these muscles differ in several aspects. The SOL exhibits oxidative metabolism and is mainly composed of type I slow-twitch fibers, while the GAS has a mixed oxidative–glycolytic metabolism and is predominantly composed of type II fast-twitch fibers [22]. These differences explain the distinct functions of these two muscles, with the SOL playing a role in maintaining balance as a postural muscle and the GAS being involved in rapid flexion movements of legs and ankles. When comparing FNDC5 expression between these two muscles in basal conditions, we observed a significant expression in the GAS compared to the SOL, consistent with results reported in mice [42]. To assess how these two muscles respond to EX, we further conducted FNDC5 expression analysis after EX at different intensities. While no significant change in FNDC5 expression was found in the SOL, our results indicate that FNDC5 expression in the GAS also depended on EX intensity threshold, with a significant increase observed from 50% of MAS of the EX protocol. While suggesting that FNDC5 expression could be linked to muscle metabolism, this finding is consistent with a study showing that an eccentric contraction, which places greater demands on fast-twitch muscle fibers than a concentric contraction, promotes higher irisin secretion in the bloodstream [43]. Regarding the type of muscle fibers responsible for FNDC5 expression in these muscles, co-immunolabelling experiments of FNDC5 and type I slow-twitch fibers or type II fast-twitch fibers were performed. In both muscles, our results clearly showed that expression was restricted to type II fibers, which is consistent with the predominant expression observed in the GAS. Thus, when assessing the effect of intensity on FNDC5 labelling, our data logically showed no effect in the SOL, composed almost exclusively of slow-twitch fibers, whereas intensity-dependent labelling was observed in the GAS, predominantly composed of type II fibers. Thus, beyond the type of muscle metabolism, our data demonstrate in a highly original way that the type of fibers and their proportion within the muscles control the muscular expression of FNDC5 in response to EX. Additionally, the positive association found between FNDC5 expression in the GAS and blood irisin levels supports the notion that muscles containing fast-twitch fibers serve as a source of circulating irisin. In terms of future directions, these findings will require further refinement through an analysis of the different sub-categories of type II fibers (IIa, IIb and IIx) responsible for FNDC5 expression in response to EX. Interestingly, it was also recently demonstrated, in C2C12 mouse myoblast cells, that FNDC5/Irisin might be involved in the control of myofiber type since overexpression, using an eukaryotic expression vector carrying FNDC5, increased the type IIa mRNA, while downregulation, using small interfering RNA for depletion, decreased it [44]. Finally, our data showing a correlation between FNDC5/Irisin expression in these fast-twitch fibers and circulating irisin levels, suggest that the protease responsible for irisin secretion into the bloodstream compartment is also expressed and activated in these fibers. In this regard, one study suggests that the cleavage mechanism could be dependent on a membrane secretase belonging to the desintegrin and metalloproteinase (ADAM) family, as broad-spectrum inhibition of ADAM proteases in C2C12 myotubes prevented angiotensin-II-induced increase in irisin in the culture medium [45]. Given that several subtypes of ADAM have been identified, it is possible that one of them is specifically expressed in type II fibers, being overexpressed in response to EX and responsible for the release of irisin into the bloodstream. Regarding the specific expression of FNDC5 in type II fibers, it is likely associated with the highest level of PGC-1α in those fibers, as previously observed by Gouspillou et al. (2014) [46]. PGC-1α is a transcription factor induced in skeletal muscle in response to its contraction, which is known to induce FNDC5 expression [15].

## 4. Materials and Methods

### 4.1. Animals

Experiments were carried out on 8-week-old male Wistar rats (n = 26) according to the French Department of Agriculture guidelines (APAFIS #33300) and approved by the local ethics committee (C2EA, Dijon, n°105). Rats were obtained from Janvier Labs (Le Genest Saint Isle, France), housed 5 per cage, kept under a 12/12 h light/dark cycle and provided ad libitum access to food and water.

### 4.2. Physical Exercise Protocol and Animal Groups

The experimental design is summarized in Figure 9. Following a habituation period to the experimenter and treadmill apparatus (days 0 to 6), all animals were first subjected to an incremental EX test to exhaustion on a treadmill apparatus (starting from 9 m/min and increasing by 3 m/min every 2 min, day 6) to determine their maximal aerobic speed (MAS). Refractory animals to the treadmill EX during the habituation period were excluded from the experiment (n = 2). Based on their MAS values, animals were assigned to four experimental groups: sedentary (SED, n = 6), low- (treadmill speed set at 12 m/min, EX12, n = 6), medium- (treadmill speed set at 14 m/min, EX14, n = 6), high- (treadmill speed set at 18 m/min, EX18, n = 6) intensities EX. Under these conditions, EX intensities were approximately 40, 50 and 70% of MAS for EX12, EX14 and EX18 groups, respectively. In the SED group, animals with different MAS were allocated. Rats assigned to the different EX groups were trained 30 min/day using a horizontal treadmill every morning for 7 consecutive days (days 8 to 14), while SED rats were kept in their own cage in proximity of the treadmill apparatus.

### 4.3. Collection of Samples

Immediately after the last treadmill session (day 14), all rats were anesthetized with isoflurane (Belamont, Piramal Critical Care B.V., Voorschoten, The Netherlands; induction 4–5%, maintenance 2%) to collect blood from their tail vein. Note that it is well known that irisin is a training-inducible myokine, and circulating irisin increases transiently in response to EX [47]. Nygaard et al. (2015) reported that the concentration of irisin in bloodstream peaks immediately after training. In this condition, serum irisin levels resulting from FNDC5 cleavage were measured immediately after the last session of EX [48].

After blood centrifugation (+4 °C, 15 min, 2000 rcf), serum was aliquoted and immediately frozen at −80 °C until further use. Of note, irisin ELISA assessments were performed after 6 weeks of freezing on aliquots thawed only once, just before experiment. Twenty-four hours after the last treadmill session (day 15), all rats were anesthetized with a ketamine (75 mg/kg, Virbac, Carros, France)/xylazine (8 mg/kg, Bayer, Leverkusen, Germany) mix (0.1 mL/100 g, i.p.) following premedication with buprenorphine (0.05 mg/kg, s.c., Axience, Pantin, France) 30 min before anesthesia. A 5-min transcardial saline perfusion was performed to flush out blood from the vasculature. After decapitation, the brain was extracted and hippocampus (HP) of the right hemisphere was collected, weighed and promptly frozen at −80 °C until biochemical analysis. Regarding muscle analysis, FNDC5/Irisin expression and localization were assessed in two distinct calf muscles differing in fiber type composition and metabolism: the medial gastrocnemius (GAS) and the soleus (SOL). While muscles from the right side of the body were collected, weighed and immediately frozen at −80 °C until biochemical analysis, those of the left side were used for immunohistochemistry analysis. Thus, the left GAS and SOL were immediately fixed in paraformaldehyde solution (4% PFA, 9713, VWR, Fontenay-sous-Bois, France) for 48 h, before being dehydrated using increasing ethanol gradients and embedded in paraffin (automaton ASP300, Leica, Cellimap, Dijon, France). Paraffin-embedded muscle sections were cut with a microtome (HM325, Microm Microtech, Brignais, France) in 5 µm thick cross-sections, spread on SuperFrost Plus™ (J1800AMNZ, Epredia, Neuilly-Sur-Seine, France) slides and dried overnight at 45 °C.

### 4.4. Analysis of Samples

It is necessary to note that no specific antibody allowing discrimination between FNDC5 and irisin exists; hence, the term FNDC5/Irisin is commonly used in publications. This is particularly the case in immunofluorescence analysis where the expression/localization of FNDC5/Irisin is reported. The same does not apply to the Western blotting technique, as the molecular weights of FNDC5 and irisin differ (FNDC5 ~24 kDa vs. irisin ~13 kDa) [49].

#### 4.4.1. Western Blotting

After homogenization of HP and muscle samples in 7 volumes of ice-cold lysis buffer [100 mM Tris-HCl (pH 7.4), 150 mM NaCl, 1 mM ethylene glycol tetraacetic acid, 1% triton X-100, 1% protease inhibitor cocktail (P8340, Sigma-Aldrich, St-Quentin-Fallavier, France)], the protein concentration of supernatants was measured using the Lowry method (Modified Lowry Protein Assay kit, 23240, Thermoscientific, Illkirch-Graffenstaden, France). Equal amounts of protein were loaded onto sodium dodecyl sulfate–polyacrylamide gel electrophoresis (SDS-PAGE) or TGX Stain-Free Fast Cast Acrylamide gel (7.5%, 1610181 or 12%, 1610185, BioRad, Marnes-la-Coquette, France), separated and electrophoretically transferred to polyvinylidene difluoride (PVDF) membranes (0.2 µm, 161-0177, BioRad, Marnes-la-Coquette, France) using a liquid method for BDNF transfer or a semi-liquid method for FNDC5 and MyHC IIa/IIb (Turbo Transblot, 1704150, BioRad, Marnes-la-Coquette, France). After blocking nonspecific binding sites with a 5% solution of nonfat dry milk in Tris-buffered saline (TBS, 20 mM Tris/HCl, 137 mM NaCl, pH 7.4) containing 0.1% Tween 20 (TBS-T), membranes were probed with different primary and secondary antibodies according to the target protein (Table 2). Protein–antibody complexes were visualized using the enhanced chemiluminescence Western blotting detection system [ECL 2 (80196, Thermoscientific, Illkirch-Graffenstaden, France) or ECL clarity (1705061, BioRad, Marnes-la-Coquette, France) depending on the band intensity of the target protein]. Band densities were determined using the ChemiDoc Imaging System (BioRad, Marnes-la-Coquette, France) and analyzed using ImageLab software (version 6.1.0 build 7, Standard Edition, BioRad, Marnes-la-Coquette, France). Internal standard markers were used in the brain (β-actin) and muscles (αSMA or total proteins) (Table 2). The appropriate amounts of total proteins to be analyzed were determined from concentration (increasing amounts of proteins)/response (band density) curves from two rats both belonging to a particular group (on the same gel). Two experimental groups among SED, EX12, EX14 and EX18 or two different muscles (SOL vs. GAS) of rats were simultaneously analyzed on the same gel, and gels were run in duplicate with identical amounts of protein for each sample load in the gel. Data were analyzed from a representative membrane shown above each graph.

#### 4.4.2. Immunofluorescence

Muscle slices were deparaffinized and rehydrated in various successive baths of xylene and ethanol, then washed in Phosphate-buffered saline (PBS, P5368, Sigma-Aldrich, St-Quentin-Fallavier, France). Epitope retrieval was then induced by heating slices (95 °C for 20 min) in Sodium Citrate buffer (Trisodium citrate, 10 mM, 0.1% Tween 20, pH 6.0) bath. After blocking nonspecific binding sites with TBS containing 3% of goat serum (GS) for 30 min at room temperature (RT), slides were washed with PBS and then incubated overnight at +4 °C with specific primary antibody in TBS with 1% of GS: an anti-FNDC5/Irisin to detect FNDC5 and irisin and an anti-MyHC I or anti-MyHC II to identify muscle fiber types (Table 3). Every immunofluorescence was performed with a negative control, i.e., slice without the primary antibody, to visualize eventual nonspecific background staining and tissue autofluorescence. After PBS washing, muscle slices were incubated for 60 min with fluorescent secondary antibodies: Alexa Fluor −568 or −488 in TBS containing 1% of GS. Finally, muscle sections were mounted between a slide and coverslip using Fluoro-Gel mounting medium with DAPI (FP-DT094A, Interchim, Montluçon, France). Slides were observed and acquired by using an epifluorescence microscope (Axioscop Imager.M2, Zeiss, Oberkochen, Germany). The fluorescence intensity was quantified using ZEN 2 (blue edition) software (Zeiss, 3.7, Oberkochen, Germany).

#### 4.4.3. Enzyme-Link Immunosorbent Assay (ELISA)

The levels of irisin in serum and HP were determined with a commercial ELISA kit (Irisin recombinant Elisa Protocol, EK-067-29, Phoenix Pharmaceuticals, Schiltigheim, France). The limit of sensitivity was fixed at 1.29 ng/mL. Measurements were performed according to the manufacturer’s instructions, with samples diluted at 1/10. All assays were performed in duplicate, and a positive control validated the experimental conditions.

### 4.5. Data and Statistical Analysis

GraphPad Prism 9.5.0 (Dotmatics, Boston, MA, USA) was used for statistical analysis and graph creation. Data were expressed as means ± standard deviations (SD). Based on normality and equal variance (Shapiro–Wilk test), differences were assessed using (1) a parametric *t*-test or non-parametric Mann–Whitney test for two groups, (2) an ordinary one-way ANOVA (Fisher’s LSD or uncorrected Dunn’s test) or Kruskal–Wallis test for four groups, and (3) a Pearson or Spearman test (unpaired values), or paired *t*-test or Wilcoxon test (paired values) for correlations. A value of *p* < 0.05 was considered statistically significant. The Dixon test identified extreme values as outliers, allowing their exclusion from statistical analyses. 

Graphical abstract was realized with BioRender website (Toronto, ON, Canada, accessed on 21 December 2023).

## 5. Conclusions

In conclusion, our findings indicate that the intensity-threshold-dependent increase in cerebral BDNF is associated with a similar rise in circulating irisin levels. The correlation between these two parameters observed in our study strongly implies the involvement of peripheral irisin rather than central irisin in the hippocampal elevation of this neurotrophin. Furthermore, our study underscores the role of muscles enriched with fast-twitch fibers in the expression of the precursor FNDC5 following EX. Notably, our data strongly suggest that these fast-twitch fibers are also the key contributors to the surge in circulating irisin levels. From a translational research perspective, our findings propose that protocols emphasizing the engagement of muscles with a substantial proportion of fast-twitch fibers could optimize EX-induced cognitive enhancement. Moreover, our results suggest that irisin holds promise as a peripheral marker for cognitive function, providing an avenue for objective assessment through a straightforward blood test.

## Figures and Tables

**Figure 1 ijms-25-01213-f001:**
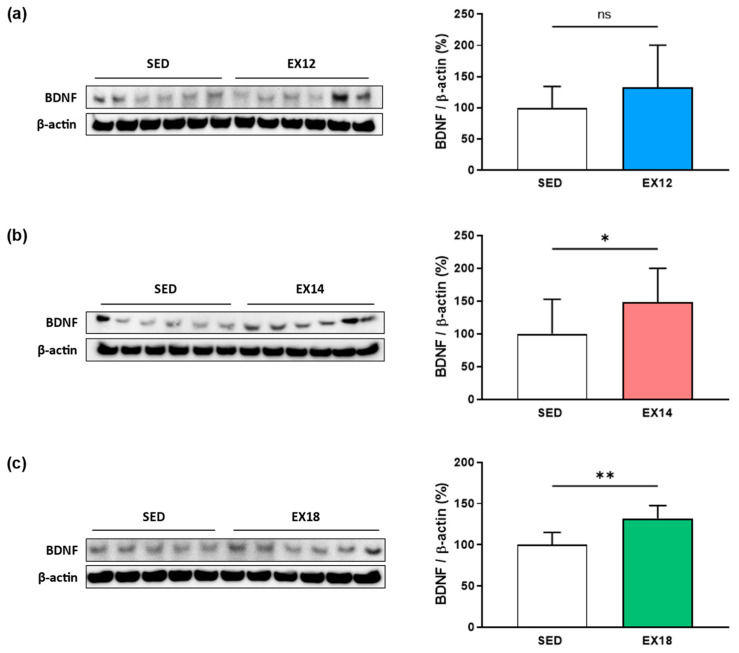
Impact of EX intensity on hippocampal BDNF expression. The EX groups were compared to the SED group (**a**) SED vs. EX12, (**b**) SED vs. EX14, (**c**) SED vs. EX18. White, blue, red and green bars were used to represent SED, EX12, EX14 and EX18 groups, respectively. BDNF (15 kDa)/β-actin (42 kDa) values (means ± SD) were expressed as % of SED. For each comparison, a representative immunoblot is shown to the left of the bar graph. Note that according to a Dixon test, an extreme value of SED was identified as an outlier, allowing it to be excluded from the statistical analyses. ns: non-significant, * *p* < 0.05, ** *p* < 0.01 compared to SED group (n = 5–6/group).

**Figure 2 ijms-25-01213-f002:**
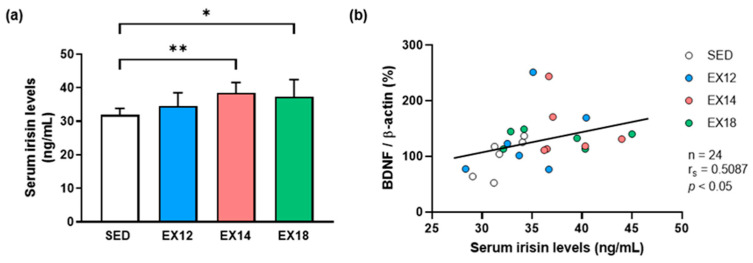
Impact of EX intensity on serum irisin levels and correlation with hippocampal BDNF/β-actin expression in SED and EX groups. (**a**) Serum irisin levels (means ± SD) were measured by ELISA test. (**b**) Scatter plot between individual levels of HP BDNF expression (% of SED values) and serum irisin levels (ng/mL). White, blue, red and green bars and scatter plots were used to represent values obtained for SED, EX12, EX14 and EX18 groups, respectively. * *p* < 0.05, ** *p* < 0.01 compared to SED group (n = 6/group).

**Figure 3 ijms-25-01213-f003:**
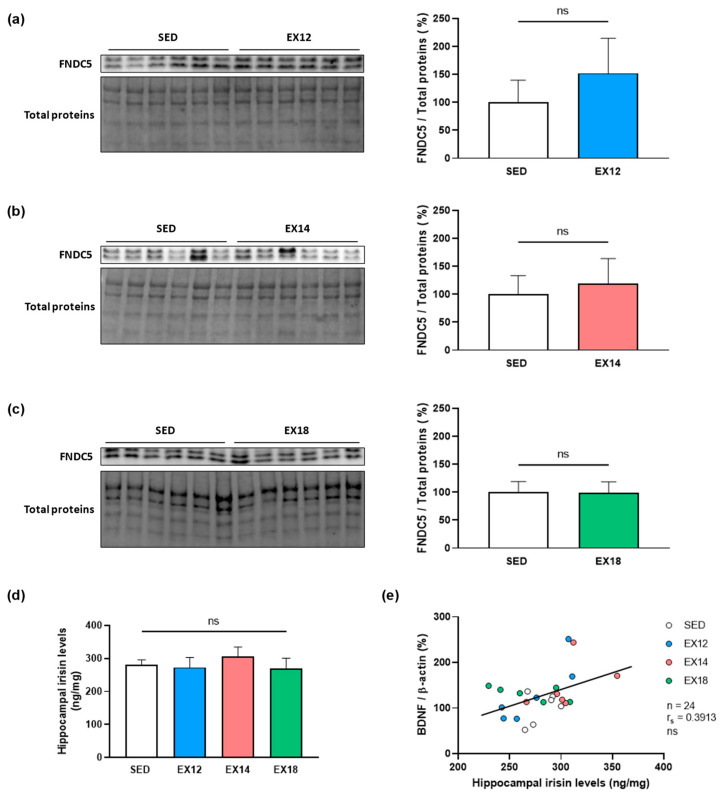
Impact of EX intensity on FNDC5 expression and irisin levels in the HP and correlation between these two parameters. The EX groups were compared to the SED group (**a**) SED vs. EX12, (**b**) SED vs. EX14, (**c**) SED vs. EX18. FNDC5 (24 kDa)/Total proteins values (means ± SD) were expressed as % of SED group. For each comparison, a representative immunoblot is shown to the left of the bar graph. (**d**) Hippocampal irisin levels (means ± SD) in SED and EX groups were measured by ELISA test. (**e**) Scatter plot of individual levels of BDNF/β-actin expression (% of SED values) and irisin levels (ng/mg) in the HP in SED and EX groups. White, blue, red and green bars or scatter plots were used to represent SED, EX12, EX14 and EX18 groups (n = 6/group), respectively. ns: non-significant.

**Figure 4 ijms-25-01213-f004:**
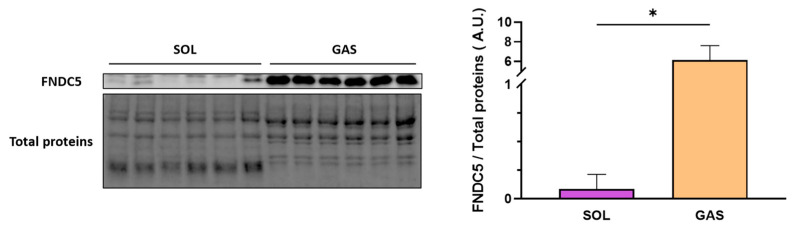
FNDC5 expression in SOL vs. GAS muscles in basal conditions. FNDC5 (24 kDa)/Total proteins values (means ± SD) obtained in SOL and GAS muscles within the same rat were expressed in arbitrary units (A.U.). A representative immunoblot is shown to the left of the bar graph. * *p* < 0.05 (n = 6).

**Figure 5 ijms-25-01213-f005:**
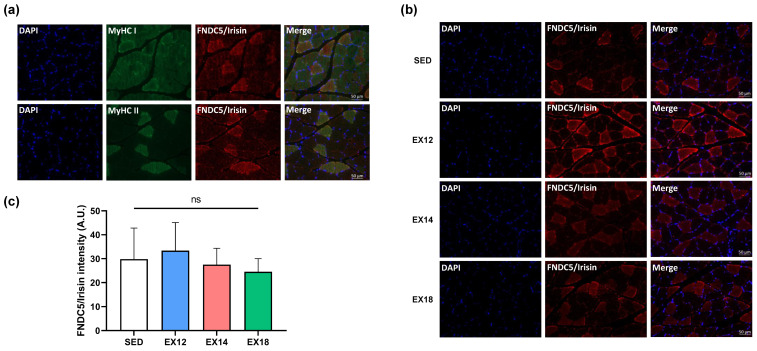
FNDC5/Irisin staining in type I and II fibers in SOL muscle. (**a**) FNDC5/Irisin staining in SOL type I (MyHC I) and II (MyHC II) fibers. From all immunofluorescence data obtained for the four experimental groups (SED, EX12, EX14 and EX18) (**b**), quantification of the type II fibers FNDC5/Irisin staining mean intensity are expressed as means ± SD (**c**). DAPI (blue) was used as a nuclear marker. Green- and red-labelled immunofluorescences represented type I/II fibers and FNDC5/Irisin, respectively. Scale bar = 50 µm. On bar graph, white, blue, red and green were used to represent SED, EX12, EX14 and EX18 groups, respectively (n = 6/group). ns: non-significant.

**Figure 6 ijms-25-01213-f006:**
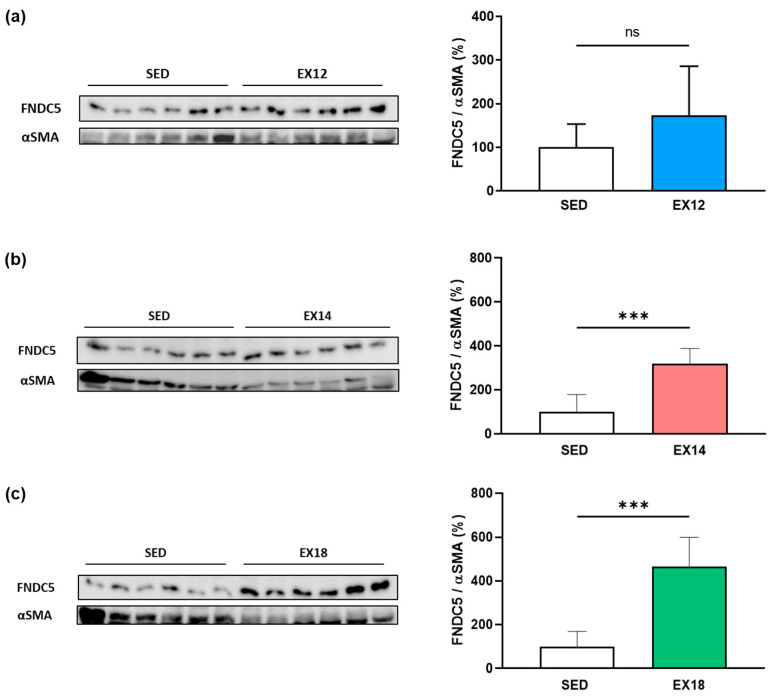
Impact of EX intensity on FNDC5 expression in GAS muscle. The EX groups were first compared to the SED group (**a**) SED vs. EX12, (**b**) SED vs. EX14, (**c**) SED vs. EX18. White, blue, red and green bars were used to represent SED, EX12, EX14 and EX18 groups, respectively. FNDC5 (24 kDa)/αSMA (42 kDa) values (means ± SD) were expressed as % of SED. For each comparison, a representative immunoblot is shown to the left of bar graph. ns: non-significant, *** *p* < 0.001 compared to SED (n = 6/group).

**Figure 7 ijms-25-01213-f007:**
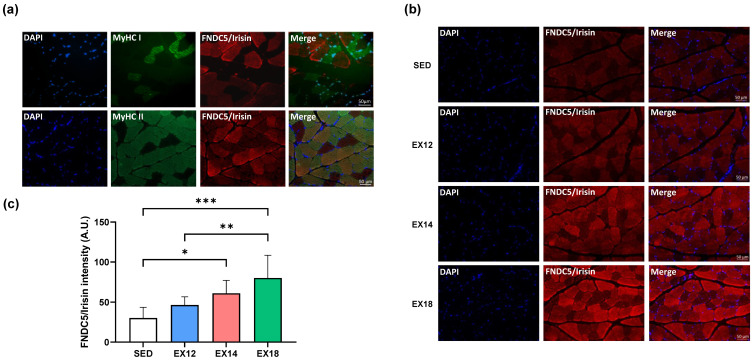
FNDC5/Irisin staining in type I and II fibers in GAS muscle. FNDC5/Irisin staining in GAS type I (MyHC I) and II (MyHC II) fibers (**a**). From all immunofluorescence data obtained for the four experimental groups (SED, EX12, EX14 and EX18) (**b**), quantification of the type II fiber FNDC5/Irisin staining mean intensity are expressed as means ± SD (**c**). DAPI (blue) was used as a nuclear marker. Green- and red-labelled immunofluorescences represented type I/II fibers and FNDC5/Irisin, respectively. Scale bar = 50 µm. On bar graph, white, blue, red and green were used to represent SED, EX12, EX14 and EX18 groups, respectively. * *p* < 0.05, ** *p* < 0.01, *** *p* < 0.001 (n = 6/group).

**Figure 8 ijms-25-01213-f008:**
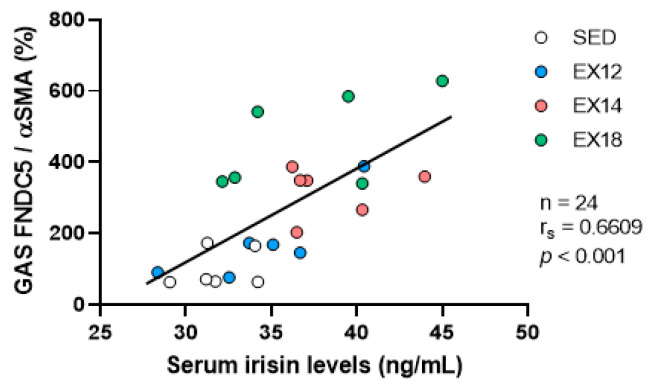
Correlation between GAS FNDC5/αSMA expression and serum irisin levels in SED and EX groups. Scatter plot between individual levels of GAS FNDC5 expression (% of SED values) and serum irisin levels (ng/mL). White, blue, red and green scatter plots were used to represent correlation values obtained for SED, EX12, EX14 and EX18 groups, respectively (n = 6/group).

**Figure 9 ijms-25-01213-f009:**
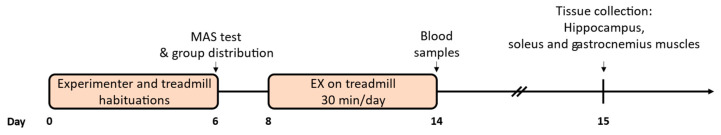
Experimental design and sample collection. After habituation and completion of the maximal aerobic speed (MAS) test, rats were allocated into groups (n = 6/group). Rats subjected to treadmill EX practiced 30 min a day during 7 consecutive days. Blood collection was performed just after the last treadmill boot, whereas the collection of tissues was carried out 24 h after.

**Table 1 ijms-25-01213-t001:** Baseline characteristics of SED and EX rats.

Baseline Characteristics/Groups	SED	EX12	EX14	EX18
Body weight (BW, g)	373.30 ± 22.51	385.00 ± 17.61	390.80 ± 15.30	375.00 ± 11.40
Soleus muscle weight (SOLW, mg)	153.70 ± 25.14	180.10 ± 17.05	188.40 ± 14.49 *	181.50 ± 26.13
Gastrocnemius muscle weight (GASW, mg)	654.70 ± 171.90	772.10 ± 108.70	813.20 ± 88.90	760.60 ± 85.53
SOLW (mg)/BW (g)	0.41 ± 0.07	0.47 ± 0.04	0.48 ± 0.04	0.48 ± 0.07
GASW (mg)/BW (g)	1.75 ± 0.41	2.01 ± 0.29	2.08 ± 0.19	2.03 ± 0.23
Average MAS (m/min)	27.00 ± 11.20	32.00 ± 1.60	30.00 ± 2.70	27.00 ± 5.00

Body (BW), soleus muscle (SOLW) and gastrocnemius muscle (GASW) weights were measured in sedentary (SED) and exercised (EX12, EX14 and EX18) rats. From these values, the ratios SOLW/BW and GASW/BW were assessed. BW was measured on the day of the last boot of EX (J14, see Materials & Methods Section 4.2. Physical Exercise Protocol and Animal Groups), whereas SOLW and GASW were determined immediately after sampling (J15). For each group, the average of maximal aerobic speed (MAS) was determined at the end of the incremental test, just before the constitution of rat groups (J6). Values are expressed as means ± SD. * *p* < 0.05 (n = 6/group).

**Table 2 ijms-25-01213-t002:** List of primary and secondary antibodies used for Western blotting technique.

Target Proteins, MW	Primary Antibody	Secondary Antibody
FNDC5, 24 kDa	Abcam ab174833, rabbit monoclonal, 1/3000 + nonfat dry milk 5%	Jackson ImmunoResearch Lab 111-035-144, HRP-conjugated goat polyclonal, 1/25,000
BDNF (mature form), 15 kDa	Abcam ab108319, rabbit monoclonal, 1/3000 + nonfat dry milk 5%	
αSMA (muscle internal control), 42 kDa	Abcam ab28052, mouse monoclonal, 1/5000 + nonfat dry milk 5%	Jackson ImmunoResearch Lab 115-035-166, HRP-conjugated goat polyclonal, 1/30,000
β-actin (hippocampal internal control), 42 kDa	Sigma-Aldrich A5441, mouse monoclonal 1/5000 + nonfat dry milk 5%	

Fibronectin type III domain containing 5 (FNDC5), Brain-derived neurotrophic factor (BDNF), α-Skeletal muscle actin (αSMA).

**Table 3 ijms-25-01213-t003:** List of primary and secondary antibodies used for the immunofluorescence technique.

Target Proteins	Target Cells	Primary Antibody	Secondary Antibody
FNDC5/Irisin	\	Abcam ab174833, rabbit monoclonal, 1/150	A11036 Invitrogen, Alexa Fluor 568, 1/800
MyHC I	Slow muscle oxidative fibers	Abcam ab11083, mouse monoclonal, 1/200	A11029 Invitrogen, Alexa Fluor 488, 1/1000
MyHC II	Fast muscle glycolytic/oxidative fibers	Abcam ab51263, mouse monoclonal, 1/300	

Fibronectin type III domain containing 5 (FNDC5), Myosin heavy chain (MyHC).

## Data Availability

The datasets used and/or analyzed during the current study are available from the corresponding author on reasonable request.

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
