# Peer review of "Impact of Exercise Intensity on Cerebral BDNF Levels: Role of FNDC5/Irisin"

_ijms, 2024, doi:10.3390/ijms25021213_

Round 1
Reviewer 1 Report
Comments and Suggestions for Authors
The manuscript "Impact of exercise intensity on cerebral BDNF levels: role of FNDC5/Irisin" is interesting and highlights how the beneficial effects of physical exercise involve more than ever a whole series of factors that relate muscle and CNS. In this case in particular BDNF and Irisin.
However, some critical issues must be resolved before the manuscript can be accepted for publication.
1) A particular limitation to the study is the fact that blood samples were obtained immediately after the motor protocol with all remaining sampling 24 hours later. This temporal gap could create a difference in terms of cellular adaptation in the CNS that could mask/modify the effects induced by exercise and which would no longer be correlated. The authors should highlight and justify this limitation.
2) In lines 336-343 the novelty of the work is the fact that irisin would be more expressed/released from type II fibers. The authors should expand this part by highlighting or proposing the possible molecular mechanisms involved in response to increasing intensities of exercise.
3) The authors assume that they can discriminate between the 13 and 24 KDa bands and that there are no additional non-specific bands. From the images there are some doubts about it. Furthermore, alpha sarcomeric actin is a fairly clean antibody and from the images it appears to be replicated from gels already analyzed with very confusing images.
4) The Elisa Irisin kIt assay is much criticized, especially because it does not detect in samples stored for too long. Were the samples tested fresh? How long were they stored in the fridge/freezer? This entails a responsibility of the authors towards other researchers working with these kits.
5) In figures 1 and 6 the SED values are normalized to 100. But why were different SED groups used? The bands are really different.
6) In figure 4 the significance of 0.0313 seems highly underestimated. Check the statistics again.
7) Lines 29-31 add reference.
8) Line 48 add reference on recent work with muscle-CNS crosstalk.
9) Line 30: Cancers? In what sense? Delete or explain.
Comments on the Quality of English LanguagePlease always consider a final review by a native English speaker.
Reviewer 2 Report
Comments and Suggestions for Authors
This paper investigates the mechanisms of EX on brain BDNF expression, they find HP BDNF expression correlated with serum irisin not brain irisin. The serum irisin level correlated with GAS muscle irisin. Irisin are predominately expressed by type 2 fast twitch muscle fiber not type 1 muscle fiber. The study is overall interesting. Data are clearly presented. Following are some suggestions to revise the manuscript before it can be accepted.
Line 86, “Whatever” may be better use “regardless”.
Figure 3 Western blot bands are two small to see. Authors should not crop too much and enlarge western blot images. Quantification histogram can be placed on the right side while western blot images on the left side for each panel of western blot images. Including the total protein blot and placed below target protein FNDC5.
Figure 4 Western blot images should not be cropped too narrow and cannot be seen well, total protein control should also be shown for entire blot. Place Blot image on the left, quantification on the right. SOL and Gas total protein pattern also do not look the same from original images and should be included.
2.5.1 should be “FNDC5 expression in SOL muscle after different extent of EX”.
2.5.2 Should be “FNDC5 in GAS muscle after different extent of EX”.
Line 206, “Vairiation” Should be “change” or “Increase” since it showed a trend of increase.
Line 312, “Variation” Should be “change”. Variation usually indicated in the same group or baseline while changes is often used for intervention mediated up or down regulation of certain parameters.
Line 325, “Above the type of muscle metabolism”. I am not clear the meaning authors wanted to convey here. Please use an alternative word to replace “above”.
Method sections, it is not clear, how many days the rats got different exercise before sacrifice because authors said trained for 7 consecutive days. The scheme looks like the rats trained for 7 days (habituation) and then exercise for 7 days. Please revise.
Comments on the Quality of English LanguageEnglish is overall good. Few need to be revised, see comments.
